# Optical Vibration Sensing Bionic Vector Hydrophone Based on Mechanically Coupled Structure

**DOI:** 10.3390/mi16111196

**Published:** 2025-10-22

**Authors:** Jinying Zhang, Jianyu Peng, Xianmei Wu, Yifan Shi, Wenpeng Xu, Yiyao Wang, Rong Zhang, Ziqi Li, Bingwen An

**Affiliations:** 1School of Optics and Photonics, Beijing Institute of Technology, Beijing 100081, China; pengjianyu101202@163.com (J.P.);; 2Yangtze Delta Region Academy of Beijing Institute of Technology, Jiaxing 314001, China; 3National Key Laboratory on Near-Surface Detection, Beijing 100081, China; 4Institute of Acoustics, Chinese Academy of Sciences, Beijing 100190, China; 5University of Chinese Academy of Sciences, Beijing 100049, China

**Keywords:** optical sensing, bionic vector hydrophone, fly *Ormia ochracea*

## Abstract

Vector hydrophones play an extremely important role in marine exploration. How to reduce the size of vector hydrophones while improving their directional detection capability is a critical issue that needs to be addressed. The auditory organ of the fly *Ormia ochracea* represents a prime example of achieving high-resolution directional detection within a compact size range. This paper proposes a vector hydrophone that integrates an *Ormia ochracea* fly-inspired mechanically coupled structure with an optical fiber vibration sensing structure, offering advantages of small size and strong electromagnetic interference immunity. The hydrophone demonstrates a good response to acoustic pulse trains and can accurately demodulate acoustic waves from 1 kHz to 10 kHz. Directional response experiments show that this hydrophone can significantly amplify the time delay differences of incoming acoustic waves. At an acoustic frequency of 9.25 kHz, the time delay amplification factor reaches approximately 50 times within the range of −90° to +90°, exhibiting good cosine directionality.

## 1. Introduction

Marine exploration is crucial for the marine economy [1] and national defense security and has significant application value in fields such as deep-sea mineral exploration, energy detection [2,3], and marine environmental monitoring [4]. Acoustic waves are the primary information carrier in marine exploration. By detecting acoustic waves, vector hydrophones can determine both the direction of incoming sound and identify the source type, playing an extremely important role in marine exploration. Researchers have been dedicated to developing various vector hydrophones [5,6] in order to meet different application requirements. However, in practical underwater applications, sensor systems often face stability issues such as temperature drift and noise interference. The literature indicates that utilizing the thermal locking effect can achieve stable system control and effectively suppress noise and drift [7], which provides a new technical approach for solving the temperature drift problem of underwater sensors, while the fabrication of compact vector hydrophones with enhanced directional detection capabilities continues to be an important research area.

Marine organisms, insects, mammals, and other animals in nature have gradually evolved various types of auditory organs through hundreds of millions of years of evolution, exhibiting remarkable characteristics in miniaturization and high resolution. These provide inspiration and reference pathways for the development of vector hydrophones [5,6]. Among these, the mechanically coupled auditory organ of the fly *Ormia ochracea* represents a prime example of achieving high-resolution directional detection of incoming waves within a compact size range. The fly *Ormia ochracea* specifically parasitizes crickets that emit acoustic waves at a frequency of approximately 4.8 kHz. As early as 1995, Miles et al. explained the sound source localization principle of the fly *Ormia ochracea*’s ears and proposed an equivalent mechanical model [8]. Inspired by this principle, researchers have designed and developed a series of bio-inspired vector acoustic sensors based on the fly *Ormia ochracea*’s ear mechanism. In 2014, Kuntzman et al. proposed an *Ormia ochracea* fly-inspired vector acoustic sensing structure based on SOI wafers, which used piezoelectric thin film detection and achieved good cosine directionality at 2 kHz [9]. In 2016, Wilmott et al. designed and fabricated a capacitive sensing *Ormia ochracea* fly-inspired structure, using two vector acoustic sensors to resolve the left–right ambiguity problem [10]. In 2018, Yansheng et al. proposed an asymmetric bio-inspired structure that achieved good cosine directionality within two resonant frequency bandwidths [11]. From 2019 to 2021, Rahaman et al. improved the sound pressure sensitivity, signal-to-noise ratio, and directionality of the sensing structure by optimizing the bio-inspired structure [12,13,14,15,16]. From 2022 to 2025, Rahaman et al. further improved the sensing structure to achieve 3D sound source localization [17,18]. In 2025, Yi et al. proposed a compact structure with dual-membrane coupling, with the designed sensing structure exhibiting good cosine directionality below 2 kHz [19]. The above *Ormia ochracea* fly-inspired vector acoustic sensors employ electrical vibration sensing methods such as piezoelectric or capacitive approaches, which can conveniently detect sound direction in air but require solutions for electrical insulation and electromagnetic shielding issues in marine detection applications. Fiber-optic vibration sensing uses optical signals as vibration carriers, possessing inherent electromagnetic interference immunity, making it highly promising for marine detection applications.

In recent years, ocean soundscape monitoring has received widespread attention as an important means of assessing the health status of marine ecosystems and the impact of human activities [20]. Fiber-optic acoustic sensing technology has become an ideal choice for underwater monitoring due to its cost-effectiveness, environmental sustainability, and capability for real-time coverage over extensive distances. In 2009, H.J. et al. from the University of Maryland proposed using a low-coherence fiber interferometric system to construct a Fabry–Perot interferometer for detecting micro-vibrations in *Ormia ochracea* fly-inspired sensing structures [21], achieving significant phase difference amplification at 10 kHz with an amplification factor of approximately 8. In 2024, Xin et al. developed a compact low-frequency sound source localization bio-inspired structure, achieving a directional error of ±7° within the range of −70° to 70° for acoustic waves below 1 kHz [22]. While the above *Ormia ochracea* fly-inspired sensing structures all employ fiber-optic vibration sensing, their interference-based phase modulation sensing structures essentially detect changes in optical path difference, which is influenced by multiple factors, making acoustic wave detection susceptible to external interference. The DFB-FL (distributed feedback fiber laser) optical vibration sensing structure based on wavelength modulation detects acoustic waves by monitoring changes in optical wavelength, which are primarily influenced by fiber Bragg grating strain, offering superior anti-interference capability for acoustic wave detection. However, current DFB-FL-based hydrophones are mainly scalar hydrophones [23,24], and velocity or acceleration vector hydrophones [25], with no reported applications of DFB-FL optical vibration sensing structures in *Ormia ochracea* fly-inspired sensing structures. This paper proposes an *Ormia ochracea* fly-inspired vector hydrophone based on DFB-FL optical vibration sensing, which has excellent application potential in marine exploration fields such as underwater target identification, fish migration monitoring, and security surveillance.

## 2. Principle and Methods

### 2.1. High-Sensitivity Sensing of DFB-FL Based on Phase-Shifted Fiber Bragg Grating

The principle of DFB-FL based on a phase-shifted fiber Bragg grating is shown in Figure 1: A 980 nm single-mode laser is applied to an erbium-doped *π*-phase-shifted fiber Bragg grating. Due to stimulated emission, a 1550 nm broadband laser is excited in the fiber. Subsequently, during the continuous reflection process of this broadband laser in the fiber Bragg grating, a phase shift occurs at the phase-shift point before transmission. Due to the characteristics of the phase-shifted fiber Bragg grating, a narrow linewidth transmission window is generated in the output laser spectrum, thereby achieving narrow-band laser output. The center wavelength *λ_B_* of the output light satisfies the grating equation:(1)$$\lambda_{B}=2n_{eff}\Lambda$$
where *n_eff_* is the effective refractive index of the fiber core and Λ is the grating period of the fiber Bragg grating.

When external acoustic pressure acts on the fiber Bragg grating, due to the effect of acoustic pressure *P*, it can be approximately considered that the fiber Bragg grating is subjected to isotropic stress. Among these, the axial stress *ε* applied to the fiber Bragg grating plays a dominant role, and this stress is given as follows [26]:(2)ε=−P(1−2v)/E
where *E* and *v* are the Young’s modulus and Poisson ratio of the coating, respectively.

The wavelength of a DFB-FL is denoted as *λ_B_*, and the change in wavelength, Δ*λ_B_*, is related to the axial strain *ε* applied to the grating by the following [27]:(3)$$\Delta \lambda_{B}=\lambda_{B}(1-P_{e})\varepsilon$$
where *P_e_* is a physical parameter that depends only on the fiber itself, with an approximate value of 0.22 [27].

Substituting Equation (2) into Equation (3) yields Equation (4):(4)$$\Delta \lambda_{B}=[-0.88\lambda_{B}(1-2v)/E]P$$

Using a M-Z (Mach–Zehnder) demodulation system to detect the center wavelength of the DFB-FL output light, the relationship between the interference light phase φ and the optical wavelength *λ_B_* is as follows:(5)$$\phi =\frac{2\pi nl}{\lambda_{B}}$$
where *n* is the refractive index of the fiber inside the M-Z demodulation system, and *l* is the arm length difference between the two if interference arms in the dual-beam interference demodulation system.

And then, the final equation describing the relationship between the interference light phase *φ* of the M-Z demodulation system and the sound pressure *P* is given by(6)$$\Delta \varphi =-\frac{2\pi nl}{\lambda_{B}^{2}}\Delta \lambda_{B}=\frac{1.76\pi nl(1-2v)}{\lambda_{B}E}P$$

Thus, it can be proven that when the M-Z demodulation system is determined and the coating material on the outer surface of the fiber Bragg grating is fixed, there exists a linear relationship between the optical phase change obtained from the output of the M-Z demodulation and the sound intensity. Therefore, the detection of acoustic pressure signals can be achieved using the above principle.

### 2.2. Ormia ochracea Fly-Inspired Time Delay Amplification Theory

The auditory organs of the fly *Ormia ochracea*’s ears exhibit two vibration modes in nature, namely rocking mode and bending mode. When the sound source to be detected is located near the rocking mode frequency, the two sides of the fly *Ormia ochracea*’s ears will exhibit maximum time delay differences and amplitude differences. However, in practice, the operating frequency of the fly *Ormia ochracea*’s ears does not need to be near the rocking mode, because when actual acoustic waves are sensed by the fly *Ormia ochracea*’s ears, both rocking mode and bending mode will be simultaneously excited. Under the superposition of these two modes, the opposite sides of the ears will still exhibit certain time delay differences and amplitude differences (the principle is shown in Figure 2). This also means that sensors fabricated based on this principle have the potential to detect a wide frequency range.

According to Figure 2, we can establish the differential equations of motion for the mechanical model [8]:(7)$$\left[\begin{array}{cc} M_{1} & 0\\ 0 & M_{2} \end{array}\right]\left[\begin{array}{c} \overset{\cdot \cdot }{X_{1}}\\ \overset{\cdot \cdot }{X_{2}} \end{array}\right]+\left[\begin{array}{cc} C_{1}+C_{3} & C_{3}\\ C_{3} & C_{2}+C_{3} \end{array}\right]\left[\begin{array}{c} \overset{\cdot }{X_{1}}\\ \overset{\cdot }{X_{2}} \end{array}\right]+\left[\begin{array}{cc} K_{1}+K_{3} & K_{3}\\ K_{3} & K_{2}+K_{3} \end{array}\right]\left[\begin{array}{c} X_{1}\\ X_{2} \end{array}\right]=\left[\begin{array}{c} F_{1}\\ F_{2} \end{array}\right]=\left[\begin{array}{c} P\cdot S\cdot e^{i\frac{1}{2}\omega \tau }\\ P\cdot S\cdot e^{-i\frac{1}{2}\omega \tau } \end{array}\right]$$
where *S* is the ear membrane area, and *τ* is the time delay difference between the two ear membranes. Due to the theoretical symmetry of the two ears, *K*_1_ = *K*_2_ = *K*, *C*_1_ = *C*_2_ = *C*, and *M*_1_ = *M*_2_ = *M*. From this, the transfer functions between *X*_1_, *X*_2_ and the sound pressure *P* can be obtained as(8)$$H_{x_{1}P}=\frac{S(K_{3}+i\omega C_{3})(e^{i\omega \tau /2}-e^{-i\omega \tau /2})+S(K+i\omega C-M\omega^{2})e^{i\omega \tau /2}}{{\left(K+i\omega C+K_{3}+i\omega C_{3}-M\omega^{2}\right)}^{2}-{\left(K_{3}+i\omega C_{3}\right)}^{2}}$$(9)$$H_{x_{2}P}=\frac{S(K_{3}+i\omega C_{3})(e^{-i\omega \tau /2}-e^{i\omega \tau /2})+S(K+i\omega C-M\omega^{2})e^{-i\omega \tau /2}}{{\left(K+i\omega C+K_{3}+i\omega C_{3}-M\omega^{2}\right)}^{2}-{\left(K_{3}+i\omega C_{3}\right)}^{2}}$$

According to far-field conditions, the time delay *τ* in water can be obtained as(10)$$\tau =\frac{d\sin(\theta )}{1500m/s}$$
where *d* is the center-to-center distance between the two ears membranes.

By comparing Equations (8) and (9), we can derive the transfer function between *X*_1_ and *X*_2_:(11)$$H_{x_{1}x_{2}}=\frac{(K_{3}+i\omega C_{3})(e^{i\omega \tau /2}-e^{-i\omega \tau /2})+(K+i\omega C-M\omega^{2})e^{i\omega \tau /2}}{(K_{3}+i\omega C_{3})(e^{-i\omega \tau /2}-e^{i\omega \tau /2})+(K+i\omega C-M\omega^{2})e^{-i\omega \tau /2}}$$

By substituting Equation (10) into Equation (11), after derivation we obtain the following [28]:(12)$$\sin(\theta )=1500m/s\cdot \frac{\ln\left(\frac{(K_{3}+i\omega C_{3}+K+i\omega C-m\omega^{2})H_{x_{1}x_{2}}+K_{3}+i\omega C_{3}}{K_{3}+i\omega C_{3}+K+i\omega C-m\omega^{2}+H_{x_{1}x_{2}}(K_{3}+i\omega C_{3})}\right)}{\omega d}$$

Analysis of the above equation reveals that in the idealized bending mode, *H_x_*_1_*_x_*_2_ = 1, where sin (*θ*) is always equal to zero, indicating that the fly *Ormia ochracea* ear structure has no response to the direction of incoming waves. In the idealized rocking mode, *H_x_*_1*x*2_ = −1, where the angular response sin (*θ*) of the fly *Ormia ochracea*’s ear is found to be related to the phase difference between the displacement fields of the two ear membranes in the fly *Ormia ochracea* structure, which is equivalent to being related to the time delay difference between these displacement fields. The magnitude of this time delay difference depends on material parameters of *Ormia ochracea* fly-inspired structure. This equation can serve as a basis for material selection and physical simulation, while also demonstrating the time delay amplification effect of the fly *Ormia ochracea*’s ears.

A further explanation of the time delay amplification capability of *Ormia ochracea*’s ears is illustrated in Figure 3. When the acoustic signal propagates to *Ormia ochracea*’s ears, the sound first arrives at the ipsilateral ear which responds first. Subsequently, the vibration signal from the ipsilateral ear propagates along the inter-tympanal bridge toward the contralateral ear. This propagation path plays a primary role in the time delay effect of the fly’s ears. While the contralateral ear receives the vibration signal transmitted from the ipsilateral ear, it also receives the signal transmitted through the medium (air/water). Due to the small inter-tympanal distance, the time delay between the signal from this propagation path and the ipsilateral ear is extremely small. Ultimately, the vibration of the contralateral ear manifests as the superposition of acoustic signals from both propagation paths, thereby achieving time delay amplification between the two ears.

### 2.3. Vibration Sensing Method of Vector Hydrophone

The schematic diagram of the vibration sensing scheme is shown in Figure 4. Sound waves from the source first transmit to the *Ormia ochracea* fly-inspired structure, where vibrations of the *Ormia ochracea* fly-inspired structure are then transferred to the fiber Bragg grating, causing minute deformation of the fiber Bragg grating. This deformation of the fiber Bragg grating induces small changes in the output wavelength of the DFB-FL. Subsequently, the demodulation system converts these tiny wavelength changes into optical phase changes in the M-Z demodulation system, thereby establishing a correspondence between the optical phase changes in the interferometric system and the acoustic signals, and thus achieving acoustic source detection.

Stereo Lithography Apparatus (SLA) was employed to fabricate the simulated *Ormia ochracea* fly-inspired structure (Figure 5). Manufacturing errors can cause shifts in the resonant frequencies of the fabricated *Ormia ochracea* fly-inspired structure. Therefore, a laser Doppler vibrometer (LDV) was applied to test the resonant frequencies of the fabricated *Ormia ochracea* fly-inspired structure. Testing revealed that the rocking mode of the fabricated *Ormia ochracea* fly-inspired structure is located at 2840 Hz, while the bending mode is at 8360 Hz. The final manufactured vector hydrophone has an area of 3 cm^2^ and a total mass of 0.485 g. During the resonant frequency testing of the *Ormia ochracea* fly-inspired structure, the influence of the fiber Bragg gratings was neglected. Therefore, we conducted simulations using COMSOL 6.2 on the *Ormia ochracea* fly-inspired structure to compare the resonant frequencies with and without fiber Bragg gratings. In the simulation, the fiber Bragg gratings were modeled as glass cylinders with a radius of 0.5 mm and a length of 1 cm. The simulation results show that before adding fiber Bragg gratings, the rocking mode resonant frequency of the vector hydrophone was 5491.8 Hz and the bending mode resonant frequency was 5879.9 Hz. After adding fiber Bragg gratings, the rocking mode resonant frequency was 4530.3 Hz and the bending mode resonant frequency was 4833.1 Hz. This indicates that the introduction of fiber Bragg gratings causes a frequency shift of approximately 1 kHz in the *Ormia ochracea* fly-inspired structure. However, when applying the time delay amplification performance of the *Ormia ochracea* fly-inspired structure, it is only necessary to ensure that the working resonant frequency of the fabricated vector hydrophone covers the rocking mode and bending mode of the *Ormia ochracea* fly-inspired structure while staying away from excessively high-order resonant modes.

## 3. Experiment and Results

### 3.1. Anechoic Water Tank Experiment

The experiment was conducted in an anechoic water tank, with the experimental setup shown in Figure 6. This anechoic water tank can effectively eliminate acoustic reflections above 3 kHz, and the walls of the tank are lined with anechoic wedges to prevent acoustic reflections from affecting the experiment. During the experiment, both the sound source and vector hydrophone were placed underwater at a depth of 3 m below the water surface. To satisfy far-field conditions, the sound source was positioned 5 m away from the vector hydrophone.

In the experiment, pulsed acoustic signals were transmitted from the sound source, and the vector hydrophone received two channels of acoustic signals. After processing these two signals through the demodulation system, the temporal waveforms of the two channels were obtained. Through calculation, the time delay difference between the two signals can be determined, and from these time delay differences, the cosine directionality of the vector hydrophone can be derived.

### 3.2. Waveform Response to Incoming Waves

The experiment initially tested the incoming wave response of the fabricated vector hydrophone. Figure 7 shows the vector hydrophone’s response to pulse waves transmitted by the sound source. The results demonstrate that the vector hydrophone has a generally flat response to the sound source, with the resulting incoming waveforms from the vector hydrophone clearly visible in Figure 7.

From Figure 7, it can be observed that there is an amplitude difference in the directional response of the vector hydrophone at 0°. This amplitude difference mainly originates from the asymmetry produced during the manufacturing process of the fiber Bragg grating and biomimetic components, as well as during the assembly process of the fiber Bragg grating with the biomimetic components. Therefore, we did not choose to use the amplitude difference for directional determination of the incident wave. Meanwhile, it can be found that for the vector hydrophone at −50° and +50° directions, the time delay differences of the incident wave responses are completely opposite. This indicates that using the left-right ear time delay difference for directional determination of the incident wave is entirely feasible.

### 3.3. Frequency Response to Incoming Waves

Before vector performance testing, the frequency response of the vector hydrophone to incoming waves between 1 kHz and 10 kHz was tested. The tests demonstrated that this vector hydrophone has a relatively accurate acoustic frequency response. The response peaks of Channel 1 and Channel 2 at different acoustic frequencies are shown in Figure 8. It can be seen that the vector hydrophone signal response is above the noise level, indicating that the bionic optical fiber vector hydrophone can also be used as a scalar hydrophone with a wide audio frequency response.

Meanwhile, in order to demonstrate the accuracy of the acoustic frequency response of the vector hydrophone, we have listed the acoustic frequency response characteristics of the vector hydrophone in Table 1.

### 3.4. Directional Response to Incoming Waves

During the experiment, the sound source emitted 9.25 kHz pulsed acoustic signals. First, the vector hydrophone was rotated and its actual position was calibrated so that the sound source was positioned at 0° relative to the vector hydrophone. The vector hydrophone was then rotated to −90° relative to the sound source, followed by rotation towards +90° with data recorded at 5° intervals. Time delay differences were recorded at 37 angles, with the results shown in Table 2.

Based on far-field conditions and assuming the vector hydrophone has no inter-tympanal bridge, the time delay differences received by the vector hydrophone were calculated. Comparing these calculated values with the measured values reveals that the actual vector hydrophone achieved 12-fold to 68.2-fold amplification of the received acoustic signal time delay differences. Further observation of the data reveals that the time delay amplification effect of the vector hydrophone is relatively poor within the range of −5° to +5°, and the time delay amplification effect exhibits certain fluctuations with angle at other angles. This phenomenon is primarily attributed to the non-uniform response of the biomimetic structure. This response non-uniformity arises from two main factors: the inherent characteristics of the fly *Ormia ochracea*’s ear structure [8], and geometric errors introduced during manufacturing that cause variations in the biomimetic structure's modal coupling effect at different angles. 

When the theoretical time delay values under far-field conditions are amplified by 50 times, the amplified theoretical curve approximately coincides with the measured curve (as shown in Figure 9). The RMSE between the two curves is 0.0513 and the MAE is 0.0434, indicating that the two curves are similar and further demonstrating that the vector hydrophone overall exhibits approximately 50-fold time delay amplification.

Through the time delay amplification function of the vector hydrophone, small inter-aural time delays are magnified, facilitating the establishment of a clear one-to-one mapping between inter-aural time delay and planar sound source direction. This enables planar sound source localization using the inter-aural time delay of the vector hydrophone.

To further illustrate the directional detection capability of the vector hydrophone, the minimum time delay Δ*T*_min_ and the two maximum values Δ*T*_max+_ and Δ*T*_max−_ are determined. Using the following equation, the cosine directionality is calculated, and the directional pattern of the acoustic vector hydrophone is plotted with *β* in units of dB (20lg |Δ*T*_min_/Δ*T*_max_|).(13)$$\beta =20\mathrm{lg}\left|\Delta T_{\min}/\Delta T_{\max}\right|$$

The cosine directivity pattern of the vector hydrophone (Figure 10) also demonstrates that the vector hydrophone exhibits good symmetry within the angular range of −90° to +90°.

Limited by structural constraints, this vector hydrophone provides meaningful and accurate cosine directionality only in the −90° to +90° range (90° to 270° sector).

## 4. Conclusions

The *Ormia ochracea* fly-inspired mechanically coupled structure can significantly amplify time delays. Based on theoretical derivation of acoustic delay amplification, this paper optimizes the design of an *Ormia ochracea* fly-inspired mechanically coupled structure and integrates it with a DFB-FL vibration sensing structure, resulting in a vector hydrophone with a total area of 3 cm^2^ and a mass of 0.485 g. Anechoic water tank experimental results demonstrate that the hydrophone exhibits good response to acoustic pulse trains and can accurately demodulate acoustic waves from 1 kHz to 10 kHz. Directional response experiments show that this hydrophone can significantly amplify the time delay differences of incoming acoustic waves. At an acoustic frequency of 9.25 kHz, the time delay amplification factor varies within the range of 12 to 68.2 times, with an overall time delay amplification factor of approximately 50 times within the range of −90° to +90°, exhibiting good cosine directionality. The comparison between the vector hydrophone designed in this paper and the biological prototype of the fly *Ormia ochracea* is shown in Table 3. This table further illustrates the realization of the time delay amplification mechanism of the fly *Ormia ochracea* in the vector hydrophone designed in this paper.

A-ra Cho et al. pointed out in their report titled ‘Survey of Acoustic Frequency Use for Underwater Acoustic Cognitive Technology’ that the acoustic frequency range used for underwater communication is 2.5–78 kHz. The SBPs (Sub-Bottom Profilers) manufactured by iXBlue operate at acoustic frequencies of 0.5–15 kHz. The vector hydrophone fabricated in this paper can be used for the maintenance and detection of such equipment. In addition, some marine organisms (such as killer whales, long-finned pilot whales, and bottlenose dolphins) emit acoustic signals in the 1–10 kHz range, and the vector hydrophone fabricated in this paper can also be used for the observation of such marine organisms [29]. In fact, based on the time delay amplification mechanism of the fly *Ormia ochracea*, theoretically, by appropriately designing the structural parameters of the fly *Ormia ochracea* fly-inspired structure, vector detection of signals at specific frequencies can be achieved. In subsequent work, we will further design more vector hydrophones for detecting specific frequencies according to application scenario requirements, especially targeting acoustic signals below 1 kHz that are widely present in the ocean.

Due to limitations of the experimental site and experimental conditions, this paper only conducted complete testing of the time delay amplification effect of the vector hydrophone at 9.25 kHz. However, theoretically, this biomimetic structure should also exhibit time delay amplification effects at other sound frequencies. In future work, we will further test the performance of this type of vector hydrophone by improving the testing apparatus, refining experimental conditions and methods, and clarifying quantitative indicators such as calibration sensitivity curves, noise floor, and limit of detection for this type of vector hydrophone. We will also compare it with existing MEMS, piezoelectric, and fiber-optic interferometric vector hydrophones to further explore the performance limits of this type of hydrophone. Additionally, we will test the repeatability and operational stability of this hydrophone, with the expectation of deeply exploring the time delay amplification mechanism of this type of hydrophone and fully leveraging the advantages of this type of vector hydrophone.

## Figures and Tables

**Figure 1 micromachines-16-01196-f001:**
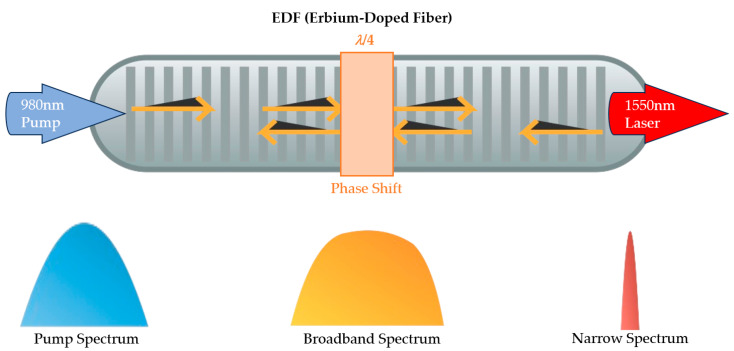
The principle of DFB-FL, where blue represents the pump light, yellow represents the broadband initial laser generated after pump excitation in the EDF, and red represents the final output laser whose center wavelength is controlled by the phase-shifted fiber Bragg grating.

**Figure 2 micromachines-16-01196-f002:**
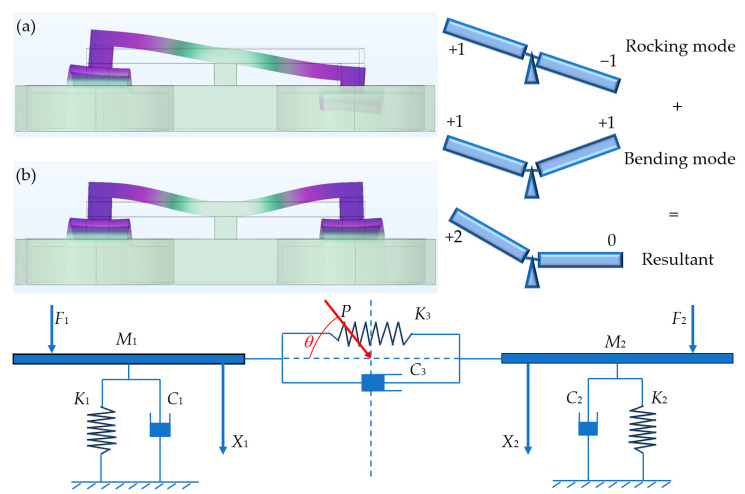
Upper left shows the rocking mode (**a**) and bending mode (**b**) of the designed *Ormia ochracea* fly-inspired structure obtained through COMSOL simulation. Upper right illustrates the vibration of the fly *Ormia ochracea*’s ears under external acoustic signals. The lower diagram shows the mechanical model of the fly *Ormia ochracea*’s ears, where *F*_1_ and *F*_2_ are the excitation forces applied to the two ear membranes by the incident sound pressure *P*, *θ* is the incident sound direction angle, *K*_1_ and *K*_2_ are the equivalent stiffnesses of the two ear membranes, *C*_1_ and *C*_2_ are the equivalent damping coefficients of the two ear membranes, *K*_3_ and *C*_3_ are the equivalent stiffness and damping of the inter-tympanal bridge, and *X*_1_ and *X*_2_ are the displacements of the two ear membranes.

**Figure 3 micromachines-16-01196-f003:**
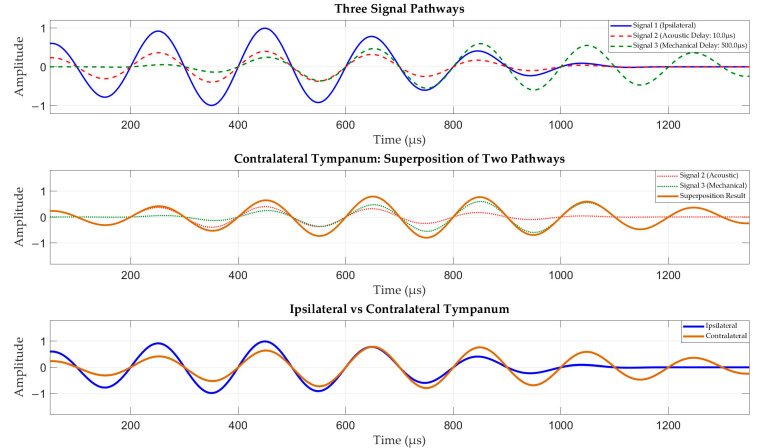
Schematic diagram of the time delay mechanism in *Ormia ochracea*. The blue solid line represents the response of the ipsilateral ear (closer to the sound source); the tan solid line represents the response of the contralateral ear (farther from the sound source); the red dashed line represents the response signal of the contralateral ear after sound propagation through water; the green dashed line represents the response signal of the contralateral ear after sound propagation through the inter-tympanal bridge.

**Figure 4 micromachines-16-01196-f004:**
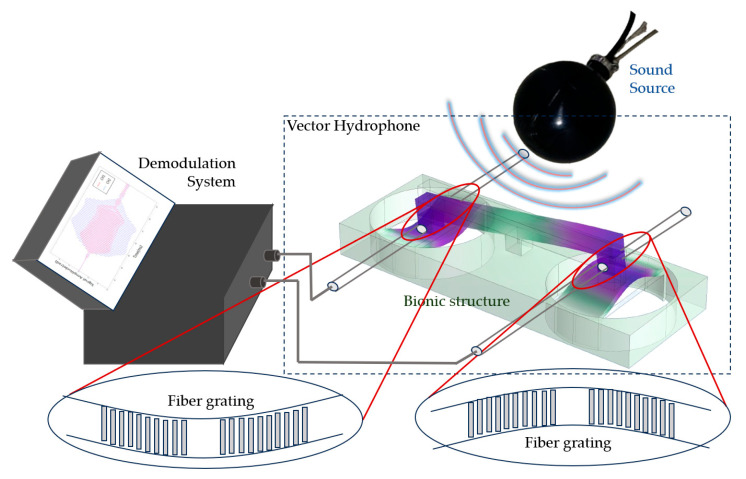
Schematic of the vibration detection scheme. Upper right: sound source; lower right: vector hydrophone consisting of bionic structure (*Ormia ochracea* fly-inspired structure) and fiber Bragg grating; left: wavelength demodulation system; bottom: deformation of the fiber Bragg grating.

**Figure 5 micromachines-16-01196-f005:**
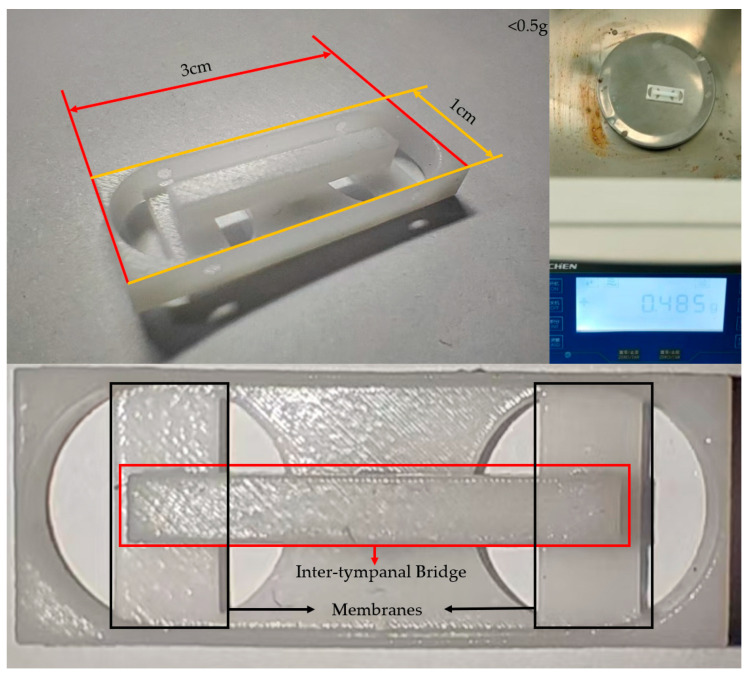
Photograph of the fabricated *Ormia ochracea* fly-inspired structure. The upper left image shows the length and width dimensions of the fabricated *Ormia ochracea* fly*-*inspired structure, the upper right shows the mass of 0.485 g, and the bottom indicates the inter-tympanal bridge and membranes of the *Ormia ochracea* fly-inspired structure.

**Figure 6 micromachines-16-01196-f006:**
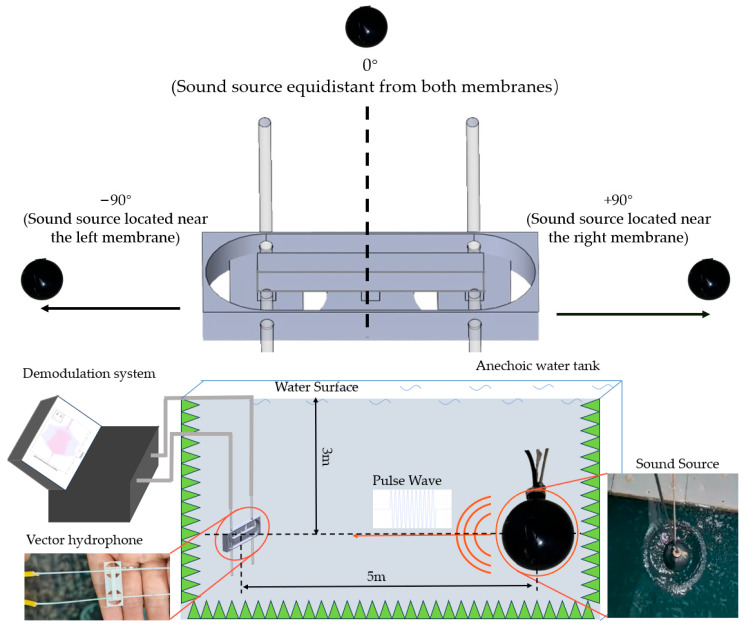
Experimental setup schematic. The lower left shows a photograph of the vector hydrophone, the lower right shows a photograph of the sound source, and the green triangles represent the anechoic wedges around the anechoic water tank. In the experiment, the negative direction is defined as the sound source being positioned close to the left membrane, while the positive direction is defined as the sound source being positioned close to the right membrane.

**Figure 7 micromachines-16-01196-f007:**
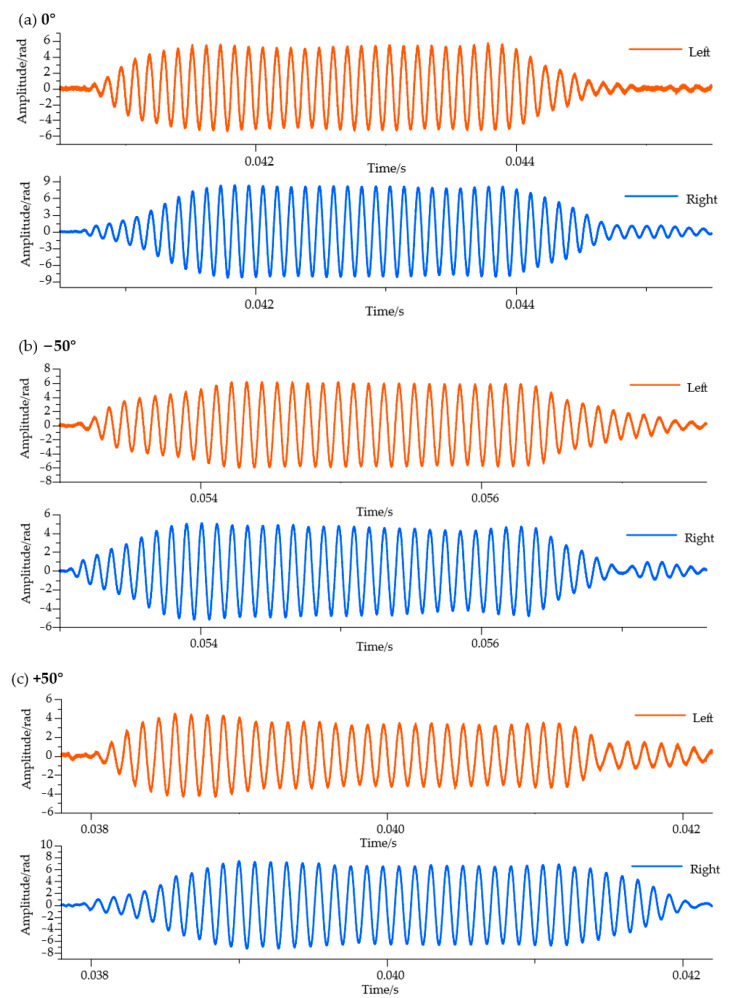
Response of the left and right ears of the vector hydrophone. Red indicates the left ear response, and blue indicates the right ear response. (**a**–**c**) show the directional response of the vector hydrophone to incident waves at 0°, −50°, and +50°, respectively. Figure (**d**) shows the signal envelope of the vector hydrophone’s directional response, where the inflection points of the signal envelope indicate the presence of signal time delay differences.

**Figure 8 micromachines-16-01196-f008:**
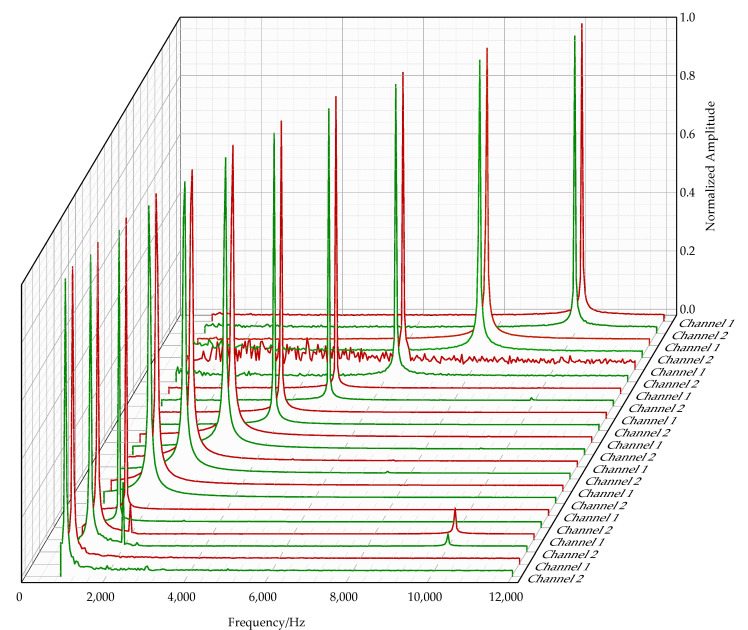
Frequency response of the vector hydrophone to incoming waves. The upper plot shows the frequency response spectra of the vector hydrophone at acoustic signal frequencies of 1 kHz, 1.25 kHz, 1.6 kHz, 2 kHz, 2.5 kHz, 3.15 kHz, 4 kHz, 5 kHz, 6.3 kHz, 8 kHz, and 10 kHz, where red represents Channel 1 and green represents Channel 2.

**Figure 9 micromachines-16-01196-f009:**
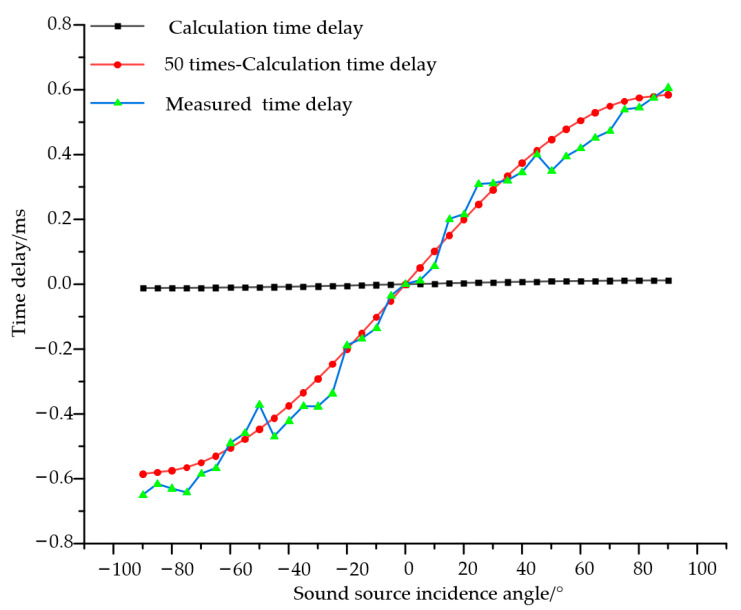
Comparison of measured and calculated time delays for the vector hydrophone. Black line: calculated time delay (approximately less than 0.01 ms overall, appearing nearly horizontal); red line: calculated time delay; blue-green line: actual time delay measured by the vector hydrophone.

**Figure 10 micromachines-16-01196-f010:**
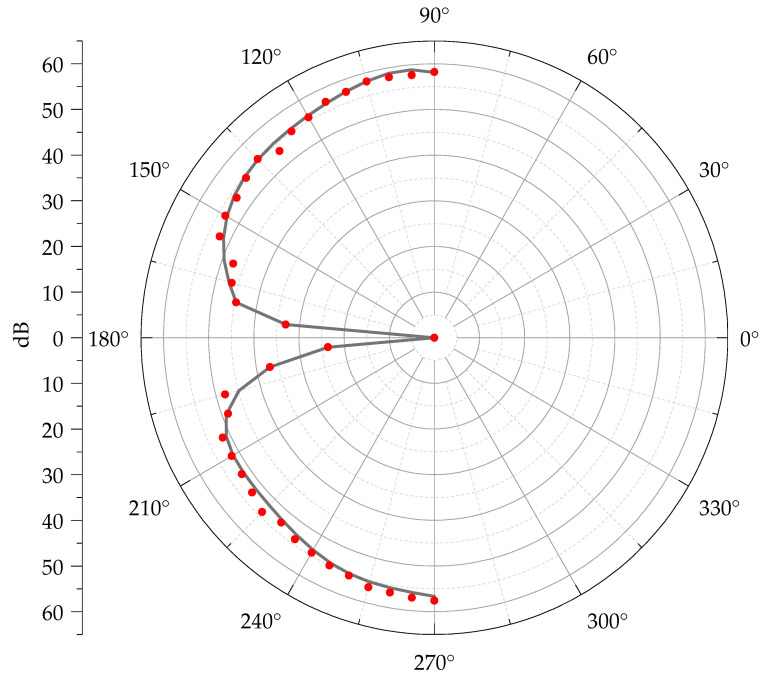
Vector hydrophone cosine directional pattern. The black line represents the fitted cosine directional curve; the red dots represent the original data points.

**Table 1 micromachines-16-01196-t001:** Acoustic frequency response of the vector hydrophone.

Sound Source Frequency/Hz	Measured Sound Frequency/Hz	Absolute Error/Hz
1000	1001	1
1250	1249	1
1600	1602	2
2000	2003	3
2500	2499	1
3150	3147	3
4000	3996	4
5000	4997	3
6300	6304	4
8000	8001	1
10,000	10,004	4

**Table 2 micromachines-16-01196-t002:** Time delay amplification of the vector hydrophone from −90° to +90°.

Angle/°	Calculation Time Delay/ms	Measured Time Delay/ms	Amplification Factor of Time Delay
−90	−1.17 × 10^−2^	−0.6496	55.7
−85	−1.16 × 10^−2^	−0.6158	53.0
−80	−1.15 × 10^−2^	−0.6302	54.9
−75	−1.13 × 10^−2^	−0.6422	57.0
−70	−1.10 × 10^−2^	−0.5844	53.3
−65	−1.06 × 10^−2^	−0.5674	53.7
−60	−1.01 × 10^−2^	−0.4898	48.5
−55	−9.56 × 10^−3^	−0.4596	48.1
−50	−8.94 × 10^−3^	−0.372	41.6
−45	−8.25 × 10^−3^	−0.469	56.9
−40	−7.50 × 10^−3^	−0.4214	56.2
−35	−6.69 × 10^−3^	−0.3762	56.2
−30	−5.83 × 10^−3^	−0.3766	64.6
−25	−4.93 × 10^−3^	−0.3362	68.2
−20	−3.99 × 10^−3^	−0.1882	47.2
−15	−3.02 × 10^−3^	−0.168	55.6
−10	−2.03 × 10^−3^	−0.1356	66.9
−5	−1.02 × 10^−3^	−0.036	35.4
0	0	0.0008	
5	1.02 × 10^−3^	0.0122	12.0
10	2.03 × 10^−3^	0.0566	27.9
15	3.02 × 10^−3^	0.2008	66.5
20	3.99 × 10^−3^	0.2164	54.2
25	4.93 × 10^−3^	0.309	62.7
30	5.83 × 10^−3^	0.312	53.5
35	6.69 × 10^−3^	0.3206	47.9
40	7.50 × 10^−3^	0.346	46.1
45	8.25 × 10^−3^	0.4008	48.6
50	8.94 × 10^−3^	0.3494	39.1
55	9.56 × 10^−3^	0.3944	41.3
60	1.01 × 10^−2^	0.4192	41.5
65	1.06 × 10^−2^	0.4518	42.7
70	1.10 × 10^−2^	0.4726	43.1
75	1.13 × 10^−2^	0.5394	47.9
80	1.15 × 10^−2^	0.5448	47.4
85	1.16 × 10^−2^	0.576	49.6
90	1.17 × 10^−2^	0.6062	52.0

**Table 3 micromachines-16-01196-t003:** Comparison between vector hydrophone and fly *Ormia ochracea* biological prototype.

Parameter	Fly *Ormia ochracea*	This Work
Area	0.288 mm^2^	3 cm^2^
Inter-membrane distance	0.45–0.52 mm	1.75 cm
Mass		<0.5 g
Directional frequency	4.8–5 kHz	9.25 kHz
Time delay amplification factor	20×	12–68.2×

## Data Availability

The data presented in this study are available on request from the corresponding author.

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
