# Peer review of "Optical Vibration Sensing Bionic Vector Hydrophone Based on Mechanically Coupled Structure"

_micromachines, 2025, doi:10.3390/mi16111196_

Round 1
Reviewer 1 Report
Comments and Suggestions for Authors
The authors present a vector hydrophone that combines a distributed feedback fiber laser (DFB-FL) optical vibration sensing structure, based on wavelength modulation, with a mechanically coupled design inspired by the ear of the fly Ormia ochracea. This integration enables significant amplification of the time delay differences in incoming sound waves, resulting in a pronounced cosine directional response. Stereo Lithography Apparatus (SLA) was employed to fabricate the Ormia ochracea-inspired structure, which significantly reduces fabrication cost and complexity compared to traditional silicon MEMS processes. The experimental results demonstrate excellent agreement with the theoretical predictions, highlighting the synergy between the high sensitivity DFB-FL and the bio-inspired structural design.
Suggested change on page 3, line 16:
1. “The wavelength of a DFB-FL is denoted as λB​, and the change in wavelength, ΔλB​, are related to the axial strain ε applied to the grating by [25]:
ΔλB​=λB​(1−Pe​)ε (3)
where Pe​ is a physical parameter that depends only on the fiber itself, with an approximate value of 0.22 [25].”
2. The definition of the sound incident angle θ may be unclear to some readers. It is therefore recommended that Figure 5 be revised to explicitly indicate the angular ranges from 0° to +90° and 0° to –90°.
Author Response
Thank you for your comments. For detailed responses, please refer to the attachment.

Reviewer 2 Report
Comments and Suggestions for Authors
Comment on the author:
This study proposes an Optical vibration sensing bionic vector hydrophone based on mechanically coupled structure, which combines bionic structure with distributed feedback fiber laser (DFB-FL) sensing technology. It has the advantages of small size, anti electromagnetic interference, and demonstrates good directionality and time delay amplification performance in underwater acoustic detection. The article has a relatively complete structure and reasonable experimental design, but there are still problems such as unclear expression, vague theoretical definitions, and insufficient data support. Therefore, the following modification suggestions are proposed.
1. The article mentions "rocking mode" and "bending mode" on page 4, and points out that in practical work, the two will overlap, thus still showing time and amplitude differences. However, the explanation of the physical image 'why the time difference can be magnified after stacking' is not intuitive enough, which is not conducive to readers' understanding of the core advantages of bionic structures. Suggest the author to use a more intuitive description or diagram based on the modal diagram in Figure 2 to supplement the explanation of how to enhance directional sensitivity after coupling the two modes.
2. On page 6, it is mentioned that the influence of fiber Bragg gratings on the resonance frequency of bionic structures can be ignored, but specific data or simulation results are not provided to support this conclusion. Suggest supplementing relevant quality comparison or modal analysis data to enhance the credibility of the argument.
3. There is a significant fluctuation in the time delay amplification factor in Table 1 on page 10 (such as from 12 times to 68.2 times), while the conclusion uniformly states it as "about 50 times". It is recommended to explain this fluctuation in the discussion section, indicating possible structural asymmetry or measurement error sources.
4. The basis for selecting 9.25 kHz as the directional test frequency in the article is not fully explained, and the time delay measurement at this frequency only displays single experimental data without providing repeatability verification (such as standard deviation or error range of multiple measurements). Suggest the author to supplement the reasons for choosing this frequency (such as whether it is close to the resonance point or the application background requirements), and add repeated experimental data to demonstrate the reproducibility and reliability of the results.
5. Additionally, some literature suggests utilizing the thermal locking effect for stable control of the system, suppressing noise and drift, (Rong Wang et al 2024 J. Phys. D: Appl. Phys. 57 055108), If the author believes that there is reference value for the problem of underwater temperature drift, it can be cited in the introduction section; There are relevant literature in the field of ocean soundscape measurement (Research. 2025; 6: Article 0089) Reference citations can be provided.
Author Response

(The authors gave the same response as above.)

Reviewer 3 Report
Comments and Suggestions for Authors
Please see attached.

Author Response

(The authors gave the same response as above.)

Reviewer 4 Report
Comments and Suggestions for Authors
This manuscript presents a novel bio-inspired vector hydrophone that integrates the mechanically coupled structure of Ormia ochraceawith a DFB fiber laser vibration sensing system. The work addresses a significant challenge in marine exploration by proposing a compact, EMI-resistant directional acoustic sensor. While the concept is innovative and the experimental results are promising, the manuscript requires substantial revisions to enhance rigor and clarity before it can be considered for publication. The following specific points should be addressed.
- The manuscript should provide a quantitative sensitivity value (e.g., in dB re 1 V/μPa or dB re rad/μPa for optical sensors) for the proposed sensor. This value must be compared with state-of-the-art hydrophones, such as piezoelectric, MEMS-based, or fiber-optic vector sensors cited in the literature;
- The noise floor of the sensor should be quantified (e.g., in dB/√Hz or μPa/√Hz) to establish its minimum detectable pressure. Additionally, the manuscript must include explicitrepeatability curves (e.g., overlapping plots of multiple trials under identical conditions) and stability curves (e.g., long-term output vs. time under fixed acoustic exposure);
- In Figure 6, the significant amplitude differences between the two curves (left and right channels) require a detailed explanation;
- It is recommended to augment Figure 6 with an illustration of the phase difference between the two curves (e.g., via a time-shift plot or polar diagram);
- Quantify frequency characteristics with sensitivity fluctuationand error bars, and replace Figure 7;
- According to Equation 10, the theoretical maximum time delay for a 3 cm membrane spacing isτmax=03m/1500m/s=0.02ms. However, Table 1 reports measured delays up to 0.6496 ms. The manuscript must clarify this discrepancy.
Author Response

(The authors gave the same response as above.)

Round 2
Reviewer 3 Report
Comments and Suggestions for Authors
None
Reviewer 4 Report
Comments and Suggestions for Authors
The authors have basically addressed the revision requirements. It is recommended to accept the present version.